# Developing evidence for building sanitation justice: A multi methods approach to understanding public restroom quantity, quality, accessibility, and user experiences

**Madison R. E. Swayne** [1] *, **Jerel P. Calzo** [2,3], **Jennifer K. Felner** [2,3], **Megan Welsh Carroll** [1]

**1** School of Public Affairs, San Diego State University, San Diego, CA, United States of America, **2** School of Public Health, San Diego State University, San Diego, CA, United States of America, **3** Institute for Behavioral and Community Health, San Diego State University Research Foundation, San Diego, CA, United States of America

* mswayne@sdsu.edu

**Data Availability Statement:** Public restroom location data and short-form assessment data are freely available at bathrooms.sdsu.edu and are

## Abstract

Access to basic sanitation is a human right and a critical environmental determinant of health. In this paper, we detail the development of three tools to investigate sanitation justice: (1) our public restroom spatial database, (2) our field assessment tool, and (3) survey of restroom access experiences. We document our process to collect these data in a consistent, health equity-driven framework. Together, these tools comprise a suite of methods for the examination of public restrooms from the macro- to the micro-level, and highlight key opportunities to promote health and well-being among restroom-reliant populations (e.g., people experiencing homelessness) by advancing sanitation justice in the built environment. With an illustrative case study, we demonstrate how methods triangulation, using the tools in concert, can provide a comprehensive assessment of basic sanitation access in a given region–San Diego, CA. We also detail how each tool can also be used separately to assess key sanitation justice and health equity questions that may be of interest to researchers, public health practitioners, policymakers, and advocates, including: (1) where do public restrooms exist (mapping)?; (2) how accessible are public restroom facilities, and what health-supportive features do they have (field assessment)?; and (3) what are the experiences of people most reliant on the available public restroom facilities (survey)? The results of our case study demonstrate that these adaptable tools can be used to provide meaningful data on and a holistic picture of public restroom quantity, quality, accessibility, and the experiences of public restroom users in a given region.

## 1. Introduction

Access to basic sanitation, including but not limited to toilets and handwashing facilities, is a human right [1,2]. Public restrooms in particular provide dignified spaces for people to fulfill basic bodily functions and are critical resources for preventing the spread of infectious diseases

provided in the manuscript. Due to ethical, privacy, and safety concerns for the respondents of our social survey, we do not share a de-identified dataset of social survey responses. People experiencing homelessness are an extremely vulnerable population. Many unhoused people experience daily the threat of criminalization simply by existing in public space. Developing trust and protecting privacy are two critical elements of conducting research with and on behalf of unhoused people and the organizations who serve them. We asked sensitive questions regarding where people go to use the restroom, related health concerns and priorities, etc. While unlikely, these data could feasibly be used to identify an individual respondent. Please contact the SDSU Institutional Review Board with any concerns: irb@sdsu.edu.

**Funding:** The authors received support from the San Diego State University (SDSU) Emergency Funding for Student Assistance with Research, Scholarship, and Creative Activities (RSCA) Award (2021); the SDSU Summer Undergraduate Research Program (2021, 2022); SDSU's Rapid Response Grants program; SDSU's Seed Grant program; and SDSU's School of Public Affairs. No grant numbers available. The funders had no role in study design, data collection and analysis, decision to publish, or preparation of the manuscript. We have added this amended Role of Funder statement to the cover letter.

**Competing interests:** The authors have declared that no competing interests exist.

including hepatitis A, shigellosis, and influenza. Unfortunately, in the United States, public restroom availability has been declining nationwide for decades as cities have neglected restroom facilities or closed them entirely [3–6]. Despite calls for increased public restroom access, many groups of people continue to suffer the public health consequences of inadequate access to restrooms. These consequences are numerous and include: urinating and defecating in undesignated places; asking stores and restaurants (sometimes unsuccessfully) for access to a restroom [7]; racial profiling, arrests, citations, and other interactions with police [8,9]; health sequelae of "holding it" including urinary tract infections, incontinence, and kidney stones; and other repeated moments of humiliation [10–12].

Availability of safe and accessible public restrooms is a critical determinant of health [2,13,14]. Prior research finds inadequate access to restrooms and other sanitation resources disproportionately impacts populations reliant on regular access and widens health inequities among some groups, such as people experiencing homelessness, people who menstruate, and people with small children [11,14–17]. For example, a lack of restroom access was cited as a key factor in the disproportionate burden of hospitalizations and deaths related to a hepatitis A outbreak in San Diego, CA among people experiencing homelessness, to whom we will also refer as unhoused [18,19]. Unhoused people may also suffer physiological effects associated with inadequate sanitation access including dehydration and urinary tract, bladder, and yeast infections [20]. Other examples of the effects of restroom inaccessibility include restroom avoidance leading to bladder-related problems and stress for women and transgender people [11,14–16].

Despite the critical role of restrooms for public health and the relationship between access to restrooms and health, there remains a limited understanding of public restroom availability and quality in concert with individual restroom user needs and experiences. While there remains no strategic policy approach to public toilet provision [21], some crowd-sourced, online databases of public restrooms have been developed to try to fill this void (e.g. Flush-Toilet Finder [22] and Toilet Finder [23]). Both Flush-Toilet Finder and Toilet Finder are web-based mobile phone applications with information about public restroom locations. While these crowd-sourced inventories capture public restrooms as identified by users, database development and validation procedures are not codified. This may introduce several data quality issues. Crowd-sourced data are subject to measurement bias in data capture where individuals have differing perspectives on what is truly "public" and what "counts" as a public restroom [24]. Additionally, without data supervision and an understanding of contributors' potential biases, there is limited potential for ex post facto database revisions posing a significant threat to data validity [25]. Other examinations of restroom availability have relied on partnerships with cities and public records requests to municipalities for information related to public toilet provision [26,27]. Cities' responses to these public records requests may undercount public restrooms. Many municipalities lack coordinated government oversight of restrooms and do not maintain a central list of public restrooms. In many cases, public restrooms are provided by various agencies including the municipal government, parks and recreation, libraries, and other entities [26]. Where records do exist, unless they are kept up to date, accuracy issues may exist [26].

We are aware of one published study that simultaneously investigated public restroom availability, quality, and impacts on people experiencing homelessness in New York City [28]. The authors highlighted insufficient public sanitation provision, particularly in low-income neighborhoods of the city, and the varied impacts on people menstruating. Other work on homelessness and bathroom access highlights the paradox of criminalizing public urination and defecation in cities with limited public restrooms and large and growing populations of

unhoused people [26]. Ultimately, the interaction of these factors creates a pressing need for additional research and advocacy to advance equity, dignity, and public health.

This article advances work to quantify restroom availability by going beyond public records requests and open-source data on public restrooms, to combine open-source, administrative, and primary data to create a spatial database of public restrooms. We then elaborate on this database with field assessments of public restrooms to verify key facility features and a survey of unhoused people to understand their experiences with restroom (in)access. In this paper, we detail the development of these three tools–our public restroom spatial database, our field assessment tool, and survey–and document our process to collect these public restroom data in a consistent, health equity-driven framework which together provide a suite of methods for the examination of public restroom availability from the macro- to the micro-level. We conclude with an illustrative case study to demonstrate how the data generated by each tool can be triangulated methodologically to provide a comprehensive assessment of the quantity, quality, accessibility, and user experiences in a given region.

## 2. Methods

In this work, we developed a suite of tools for defining and investigating sanitation justice: public restroom availability, restroom quality, and individuals' lived experiences with public restroom (in)access. The tools include (1) the development of a spatial database of public restrooms, (2) field assessment of restroom quality, and (3) a social survey of unhoused people regarding their perspectives on public restrooms and related health issues (Fig 1). While the tools are presented together, each tool could also be used separately to assess key sanitation justice and health equity questions that may be of interest to researchers, public health practitioners, policymakers, and advocates: (1) where do public restrooms exist (*mapping*)?; (2) how accessible are public restroom facilities, and what health-supportive features do they have (*field assessment*)?; and (3) what are the experiences of people most reliant on the available public restroom facilities (*social survey*)?

The tools were developed to varying degrees through engagement with our community partner, Think Dignity. Think Dignity (TD) is a San Diego-based non-profit organization working to advance basic dignity for people lacking permanent housing. Their work includes policy advocacy, legal services, and a suite of mobile services–including showers, food distribution, and legal aid–to empower unhoused people to meet their basic needs. TD has been a leading voice in public restroom advocacy in San Diego for 16 years and has identified knowledge gaps around public restroom access over the course of their advocacy work. TD staff and leadership provided guidance on initial sources of information for the mapping database and ongoing feedback during database development. Additionally, the TD team provided guidance on item development and administration of the social survey, and hosted our research team at their events so that we could interview their guests. We chose to focus on the perspectives and experiences of unhoused people because they are (1) an underserved population in many regards, especially in the area of sanitation justice; and (2) they have been disproportionately affected by recent infectious disease outbreaks (Hep-A, Shigella) and the current COVID-19 pandemic, all of which were directly or indirectly exacerbated by lack of adequate restroom access. However, we argue here that the suite of tools we propose for understanding public restroom access is adaptable to any number of focal or "priority" populations, including but not limited to women [12], gender and sexual minorities [11], pregnant people and people on certain medications [29], people who menstruate [28], commuters [30], and an array of workers, including delivery drivers [31] and transit operators [32].

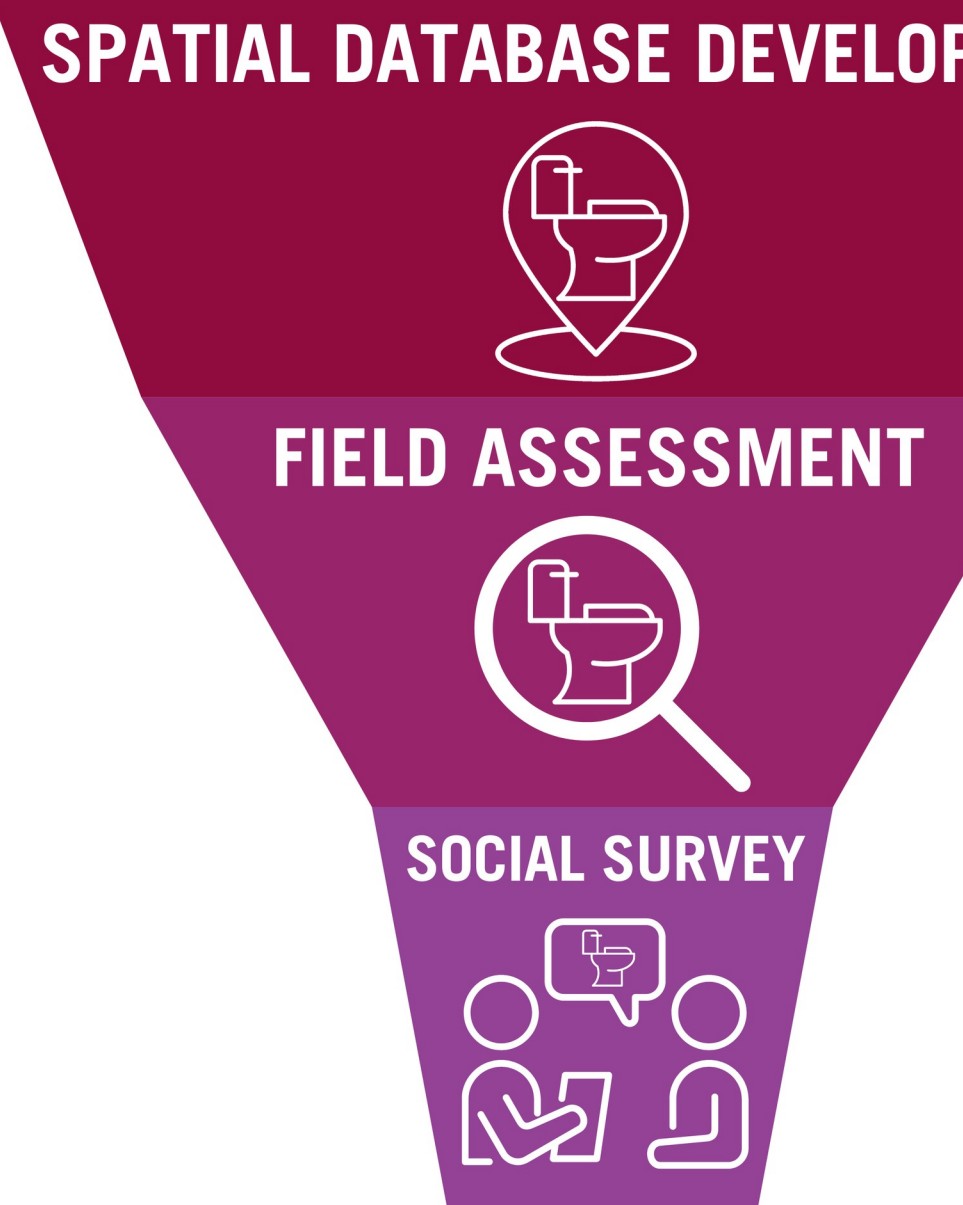

**Fig 1. Approach to documenting and understanding public restroom quantity, quality, accessibility, and other vital features.**

### 2.1 Tool 1—identification, verification, and mapping: Public restrooms spatial database development

At the most macro level of analysis, work began with identifying and mapping public restrooms in San Diego County to develop a spatial database of facilities (Tool 1: Spatial Database). Prior to this work, there was no comprehensive database of public restrooms in San Diego County.

The restroom database development process is outlined in Fig 2. Briefly, the restroom identification process started by searching organizations responsible for providing public services (parks, libraries, harbors/ports), County, and City websites for existing public restrooms lists.

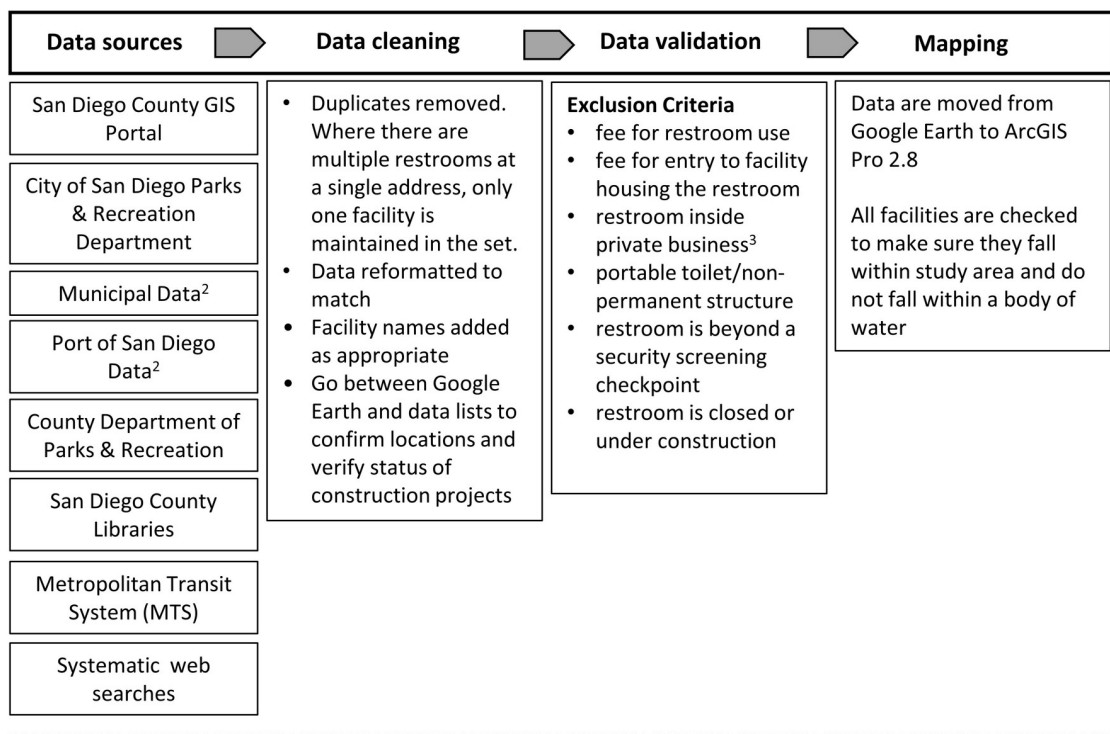

| Data sources | Data cleaning | Data validation | Mapping |
|---|---|---|---|
| San Diego County GIS Portal | • Duplicates removed. Where there are multiple restrooms at a single address, only one facility is maintained in the set.<br>• Data reformatted to match<br>• Facility names added as appropriate<br>• Go between Google Earth and data lists to confirm locations and verify status of construction projects | **Exclusion Criteria**<br>• fee for restroom use<br>• fee for entry to facility housing the restroom<br>• restroom inside private business[3]<br>• portable toilet/non-permanent structure<br>• restroom is beyond a security screening checkpoint<br>• restroom is closed or under construction | Data are moved from Google Earth to ArcGIS Pro 2.8<br><br>All facilities are checked to make sure they fall within study area and do not fall within a body of water |
| City of San Diego Parks & Recreation Department | | | |
| Municipal Data[2] | | | |
| Port of San Diego Data[2] | | | |
| County Department of Parks & Recreation | | | |
| San Diego County Libraries | | | |
| Metropolitan Transit System (MTS) | | | |
| Systematic web searches | | | |

[1]A small number of restrooms inside public businesses are included in the database. The City of Coronado, for example, administers a program to support private businesses that make their restrooms available to the public. See: https://coronadovisitorcenter.com/public-restrooms/
[2] Security screening checkpoints include those at k-12 schools, airports, and military bases

**Fig 2. Spatial database development overview.**

Where comprehensive information was not available, we contacted City or County staff and/or submitted Public Records Requests via an electronic portal.

After collating restroom information listed in these existing administrative databases in Microsoft Excel, our research team conducted a series of systematic web searches using Google Earth to further identify public restrooms across the County [33]. We conducted these searches by scanning across San Diego County and repeating searches for the terms "restroom," "public restroom," "bathroom," and "toilet." Restroom names and locations (latitude and longitude) were captured and added to the tabular dataset in Excel. These web searches were conducted during a four-month period, from October 2021 to January 2022. To eliminate duplicate restrooms, where there were multiple public restroom buildings at a facility (e.g. two bathroom buildings in a large park), individual restrooms were included only where there was the provision of both a male and female restroom and/or a gender-neutral restroom. In cases where there was a male restroom building immediately adjacent to a female restroom building, only one point was included in the dataset. Restrooms identified in the administrative records and systematic web searches were then cleaned and verified against a list of restroom inclusion criteria to ensure that all restrooms in the dataset are truly public and duplicates were removed. These inclusion criteria stipulate that each facility: cannot charge a fee for use; cannot be in a facility that charges an entry fee (paid museums, concert venues, some campgrounds); must be a permanent structure with a toilet and handwashing station (no portable toilets–they can disappear without notice); cannot be located inside of a private

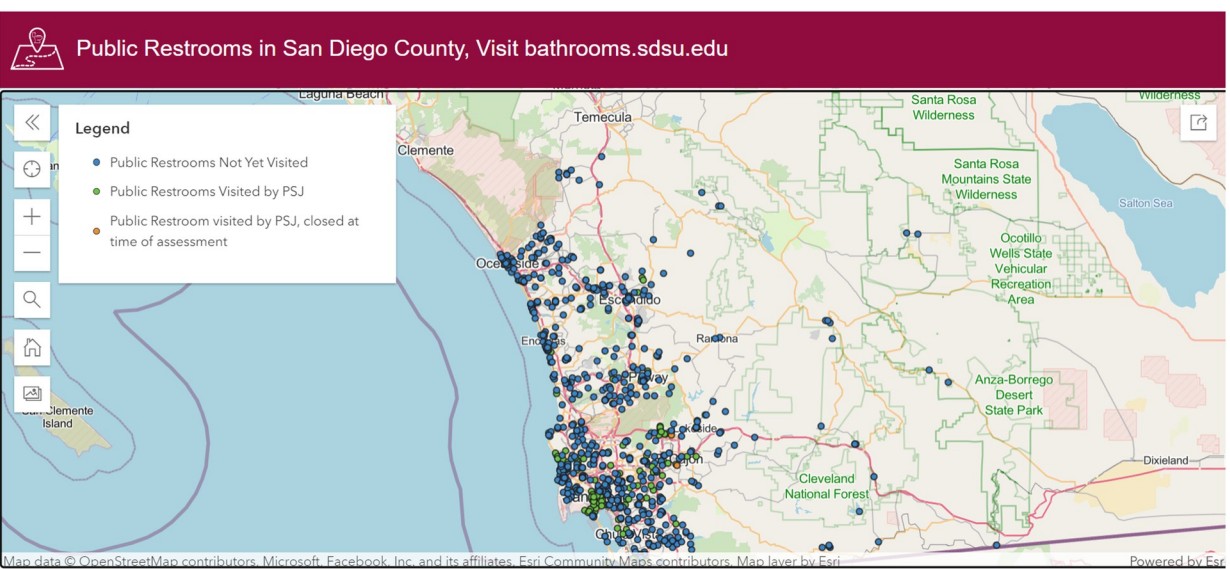

**Fig 3. Screenshot of ArcGIS Online web map of public restrooms.**

business; cannot be permanently closed or under construction; and cannot be located beyond a security screening checkpoint such as those at K-12 schools, airports, and military bases. Finally, each restroom was identified with a descriptive, unique name and a four-digit ID number.

This database development process resulted in a set of 669 public restrooms. After identification and verification of restrooms were complete, all recorded restroom coordinates were used to map restrooms as points using Esri's ArcGIS Pro version 2.8, and made public on ArcGIS Online at https://bathrooms.sdsu.edu/ (Fig 3).

## 2.2 Tool 2—field assessment of public restroom quality

Simultaneous with developing the spatial database of public restrooms, we also developed a restroom field assessment tool to gather facility-specific information on each of the identified restrooms. The field assessment tool (Tool 2) includes a long-form assessment of restroom features for sanitation and health-related features, as well as a short-form for verification of essential restroom features and location (refer to Table 1 for a comparison of restroom features included in each). The long-form assessment includes restroom features identified in previous empirical literature [17,28] and restroom access advocacy (PHLUSH [34], American Restroom Association [7]) as critical to the health of its users. The abbreviated short-form enables collection of essential information on the accessibility of a restroom while still being able to conduct spatial database verification (Tool 1) under real world constraints (e.g., when bathroom field assessments are likely to be interrupted by patrons; when gatekeepers limit assessment time, etc.). This short form, in addition to spatial location verification, includes questions on hours of operation, ADA accessibility, gender-neutral stall options, menstrual product availability, and baby changing station availability. Long-form assessment data are collected in the field using Qualtrics; short-form assessment data are collected using Esri's Field Maps mobile application. For expediency, safety, and data accuracy purposes (e.g., reliability via multiple raters), all assessment data are collected in research teams of at least two (but often three to five) people. An editable version of the long-form assessment tool is included as an appendix for readers to download and adapt (see S1 Appendix).

**Table 1. Public restroom features assessed.**

| Feature | Restroom Feature Included In: | |
|---|---|---|
| | Short-Form | Long-Form |
| Restroom Name and ID | x | x |
| Restroom Location (X-Y coordinates) | x | |
| Menstrual Product Availability | x | x |
| Baby Changing Stations | x | x |
| Gender Neutral Stalls | x | x |
| ADA Accessibility | x | x |
| Hours of Operation | x | x |
| Restroom Category<br>Categories include: public buildings (library, recreation center, pool, senior center, community center, health center, museum); outdoor facilities (park, beach, trail, campground, golf course); government facilities (Portland Loo#, government building, police/fire station, civic center); transportation (bus stop, trolley station, gas station, rest stop); commercial facilities (shopping center, restaurant); and other | x | x |
| Toilet and urinal counts | | x |
| Restroom Hygiene Features: soap, running water, hot water, toilet paper, seat covers, hand dryers, mirrors | | x |
| Restroom Security Features: stall doors, stall door locks, security cameras, security guards | | x |
| Restroom Building Features: ventilation, lights | | x |
| Adjacent features: trash cans, water fountain, bike racks, signage, transit | | x |

## 2.3 Tool 3—social survey, understanding experiences with and access to public restrooms among priority populations

In tandem with Tools 1 and 2, we developed a comprehensive social survey instrument to understand the experiences and perspectives of unhoused people, who are especially reliant on public restrooms and vulnerable to dire health consequences when restrooms are unavailable (Tool 3: Social Survey). The survey instrument, a researcher-administered structured interview, takes roughly 30 minutes to complete. The instrument was built and administered using Qualtrics. The survey asks questions about: housing status, income, and health history as it relates to toileting behaviors; how people meet basic health and hygiene needs, including restroom access and what people do when a restroom is unavailable; perceptions of important restroom features, including barriers to access and safety/security features; consequences of restroom inaccessibility, such as criminalization; and people's views on how public restroom access can be improved in San Diego.

We note that the social survey tool should be tailored depending on the population of interest. In our current work, we focus on people experiencing homelessness. Others utilizing this approach may follow our methods and include necessary domains listed above while engaging community partners and stakeholders to adapt the tool and its content to make it responsive to any unique population of interest (e.g., including domains that are relevant to the experiences of the population; ensuring the language of the tool is accessible). This tool could be tailored to other priority populations including, but not limited to, transit riders and/or operators, delivery drivers, people who menstruate, transgender and non-binary people, people with disabilities, tourists, families with small children, and people with restroom-related health concerns.

In survey item development, we received feedback from TD and drew on prior empirical scholarship in this area [17,35,36]. We recruited our social survey participant sample at TD programs including their mobile showers and food distribution events. From April through July 2022, we conducted a total of 115 researcher-administered surveys; 54 were collected in PB. All surveys were conducted face-to-face and participants were offered a $25 gift card incentive to participate. Our analysis includes PB program participants who self-identified as experiencing homelessness (n = 45); individuals who were excluded from the analysis were people who attended the events to access services but did not self-identify as currently experiencing homelessness on the survey (nevertheless, these participants were also compensated for their participation).

The San Diego State University Institutional Review Board (SDSU IRB) reviewed all research activities reported herein and deemed them a public health surveillance activity exempt from on-going review. For the survey, the research team obtained comprehensive oral informed consent from each participant, and this was witnessed by two members of the research team. In order to preserve the anonymity of participants, we did not obtain written consent (signatures). Participants received a copy of the written consent document to keep, along with a comprehensive list of local resources (e.g., for housing, food, health care, and social services).

## 3. Tools in action—A case study of pacific beach

To demonstrate the use of our tools, we present findings from a case study of Pacific Beach, a neighborhood in the City of San Diego, in San Diego County, California.

In San Diego County, sanitation justice is a chronic public health concern at the nexus of overlapping and related crises, including a longstanding and worsening housing affordability crisis [37], high and growing rates of homelessness (40.9 people experiencing homelessness per 10,000); ranked fifth nationally [38]; and repeated outbreaks of preventable diseases [39]. By one recent metric, San Diego is the least affordable housing market in the country [40]. San Diego County is the nation's fifth most populous metropolitan area and is California's second most populous county with a population of 3,298,634 people [41]. The County's boundaries include unincorporated areas and 18 cities, the most populous of which is the City of San Diego. Although City and County officials have repeatedly been warned about the potential of disease outbreaks that could ensue given the severe lack of public restrooms, action by public officials has been slow and insufficient [42,43]. San Diego faced an outbreak of hepatitis A in 2017–2018 and a Shigellosis outbreak in 2021 [44].

The Pacific Beach neighborhood (locally known as "PB") was selected for the case study because of its geographic location, population density, high tourism, and large population of people experiencing homelessness. The City of San Diego is broken up into 57 community planning areas (CPAs). These CPAs are contiguous areas generally made up of a single neighborhood and are the functional unit for land use planning in the City. We rely on CPA boundaries for our spatial analyses because of this direct connection to land use planning [45]. The PB CPA is located along the coast and is bounded by La Jolla on the north, Interstate 5 and Clairemont Mesa on the east, Mission Bay Park and Mission Beach on the south, and the Pacific Ocean on the west [46]. PB has an estimated population of 40,984 and a 2022 median household income of $91,669 [47]. The median household income in PB is slightly higher than the San Diego metropolitan area median household income of $91,003 [48]. PB is notably a community of high economic activity, including tourism. Policymakers' attention is often directed toward PB where the local, ". . .beaches and Mission Bay are vital to [the] community for both pleasure and revenue" [49].

PB is also one of the main service areas for Think Dignity's homeless services. While precise count data are unavailable for how many people are experiencing homelessness in PB, one service provider in a media report estimates that about 300 to 400 unhoused people stay in PB and nearby beach communities [50]. Overall, San Diego regional data suggest the situation is dire: homelessness has increased at least 10% in San Diego overall since the start of the pandemic, and San Diegans are falling into homelessness faster than the region can house them [51,52].

### 3.1 Mapping restrooms in Pacific Beach

The spatial database tool can describe how many bathrooms are in a region and their distribution. In addition, the database can be combined with other spatial data (e.g., land use, population and demographic statistics, health data, etc.) to address a variety of questions. We identified seven public restrooms in PB (Fig 4). Four restrooms were located along the coastline and three were inland. Relative to other CPAs, the restroom count is above average. Together, CPAs have an average of five restrooms with a minimum of zero and a maximum of 33. With its seven restrooms, PB is ranked 20th (out of 57) by number of restrooms. However, when the spatial data are integrated with other databases, such as information on encampments, it is apparent that the population pressures on these existing restrooms are severe. In 2021, 2,008 encampments were reported across the PB area over the 12-month period via the 3-1-1 phone service (also a mobile phone application called Get It Done) [53]. Although it is impossible to quantify the total number of unhoused individuals at each encampment, or the number of encampments that may exist on a given day, the total number of encampments in

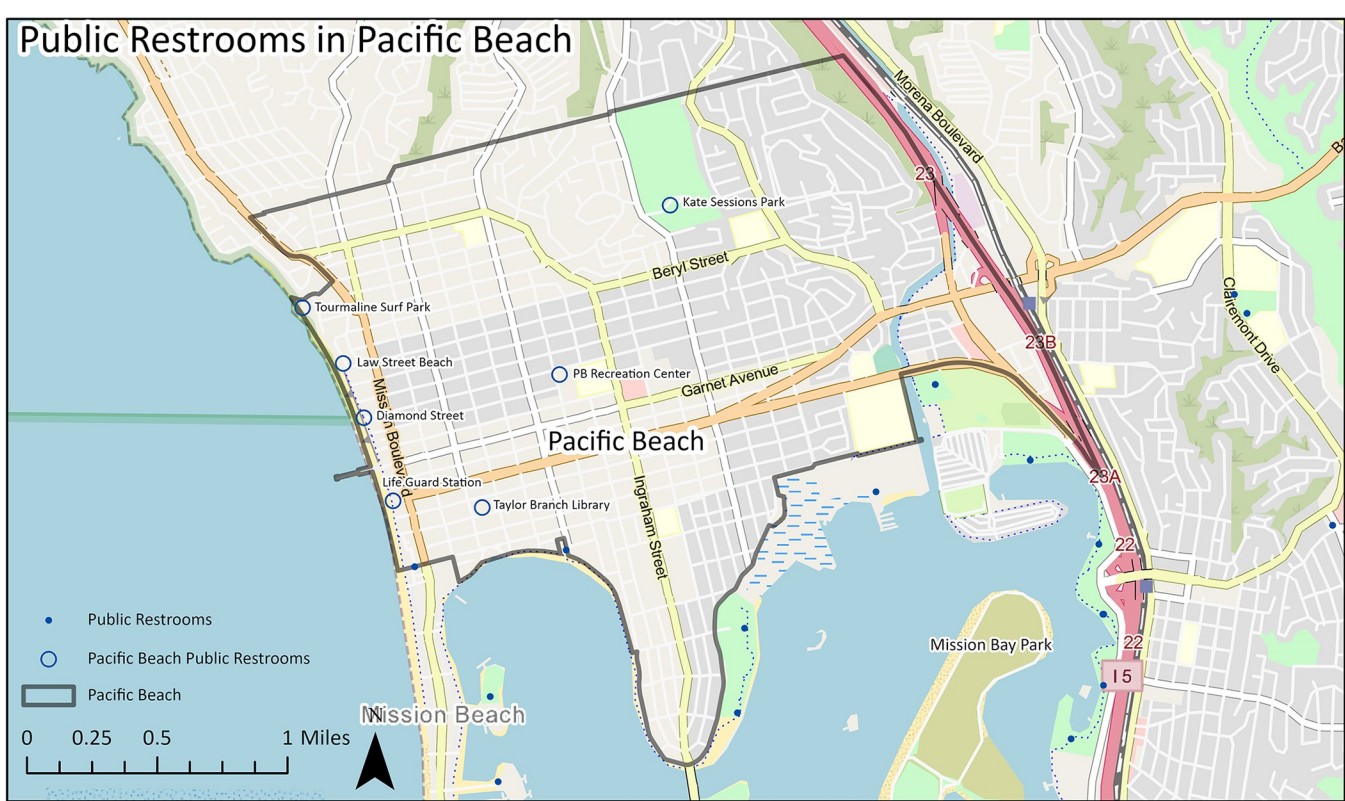

**Fig 4. Public restrooms in Pacific Beach.**

PB is much higher than the Citywide CPA average of 632 encampments. Accordingly, PB potentially has one of the highest ratios of encampments to public restrooms.

### 3.2 Field assessment of PB restrooms

In contrast to the spatial database (which primarily provides information on where restrooms are located), the field assessment tool provides information on facility type and quality. Due to the variety of types of restrooms (some in buildings and some outdoors), and the geographic distribution of the restrooms (e.g., at least two of the restrooms were an approximate 30 minute walk apart, or ≥ 1.5 miles), we visited PB on June 20, 2022 and October 11, 2022 to conduct field assessments (both long- and short-form versions of the tool were utilized).

Using the field assessment tool, we determined that five restrooms were located in outdoor facilities (four at the beach, and one at a park), and two in designated public buildings (library and community recreation center) (Table 3). During the field assessments we also attempted to confirm restroom hours of operation (Table 2). Two of the seven restrooms were advertised as being open 24/7; this could not be confirmed. At one facility advertised as open 24/7 (Lifeguard Station), a lifeguard confirmed that the adjacent public restroom is definitely not open 24/7, but estimated it is likely open from approximately 9am-10pm daily. Additionally, we could only locate posted hours at 2 of 7 facilities (both in public buildings–the Pacific Beach Recreation Center and Taylor Branch Library). Both of these facilities are closed for some or all of the weekend. Based on the field assessments, it is impossible to confirm and quantify the total hours per week during which public restrooms are available in PB.

The available features of each restroom are reported in Table 3 below. All of the public restrooms had basic features including operable toilets, functioning touch faucets, hand dryers, soap, and toilet paper at the time of the field assessment visits. Stall doors with locks were also present at all facilities. Only two facilities (28.6%) had hot water available, and two facilities had mirrors.

The availability of additional features relevant to populations underserved by public restrooms was much more limited. One facility (14.3%) had menstrual products for a fee, two facilities (28.6%) had baby changing stations, three facilities (42.9%) had a gender-neutral stall option, and five (71.4%) facilities had an ADA-accessible stall.

### 3.3 Survey of public restroom-reliant people in Pacific Beach

At the micro-level, the social survey explores perspectives of people who use public restrooms. We collected social survey data during four Think Dignity events serving the needs of unhoused people in PB: the "Street Cafe" program (3 events; n = 32 survey participants) and

**Table 3. Pacific beach restroom field assessment data.**

| Restroom Name (Facility Type) | Menstrual Products | Baby Changing Station | Gender Neutral | ADA Accessible | Soap | Hot Water | Mirror |
|---|---|---|---|---|---|---|---|
| Kate Sessions Park (Outdoor) | No | No | No | No | Yes | No | No |
| Pacific Beach Recreation Center (Public Building) | No | Yes | No | Yes | Yes | Yes | Yes |
| Taylor Branch Library (Public Building) | Yes, for fee | Yes | No | Yes | Yes | Yes | Yes |
| Lifeguard Station (Outdoor) | No | No | Yes | Yes | Yes | No | No |
| Diamond Street (Outdoor) | No | No | Yes | Yes | Yes | No | No |
| Law Street Beach (Outdoor) | No | No | Yes | Yes | Yes | No | No |
| Tourmaline Surf Park (Outdoor) | No | No | No | No | Yes | No | No |
| **Total (%)** | 1 (14.3) | 2 (28.6) | 3 (42.9) | 5 (71.4) | 7 (100) | 2 (28.6) | 2 (28.6) |

**Table 2. Pacific beach restrooms and hours of operation.**

| Restroom Facility Name | Advertised Hours | Confirmed Hours |
|---|---|---|
| Kate Sessions Park | 7am - 10pm, daily | Could not confirm |
| Pacific Beach Recreation Center | Mon 9:30am-5:30pm, Tues 9:30am-8:45pm, Weds 9:30am-3:30pm, Thurs 9:30am-5:15pm, Fri 11:30am-4:45pm, Sat 9am-2:45pm, Sun closed | Confirmed |
| Taylor Branch Library | 8am - 6pm Mon—Thu, closed Fri—Sun | Confirmed |
| Lifeguard Station | 24/7 | Could not confirm |
| Diamond Street | Not available | Could not confirm |
| Law Street Beach | Not available | Could not confirm |
| Tourmaline Surf Park | 24/7 | Could not confirm |

the "Mobile Showers" program (1 event; n = 13 survey participants). We report data below for respondents who self-identified as homeless (n = 45). Sociodemographic characteristics are summarized in Table 4. On average, participants were 51-years-old (SD = 13.1; range = 21–71). The majority were white (62%, n = 28), male (73%, n = 32), and heterosexual (93%, n = 41). Approximately 42% (n = 19) of participants reported having a physical disability. Participants reported several sources of income, including public assistance (56%, n = 25), gainful employment (18%, n = 8), recycling (16%, n = 7), and selling blood plasma (2%, n = 1). The majority reported sleeping outdoors in the past 4 weeks (60%, n = 27) and more than one-third reported sleeping in a vehicle in the past 4 weeks (36%, n = 16).

A large majority of participants indicated that they relied on public restrooms on "a typical day," underscoring the importance of regular access to these facilities. Participants provided information on the importance of accessing a toilet/urinal and where they relieved themselves on a typical day, why they used certain locations, and the easiest time to access a restroom. Nearly all participants (84%, n = 38) indicated that, on a typical day, accessing a toilet or urinal is extremely or very important; the remaining participants indicated that it was moderately or slightly important (9%, n = 4) or not at all important (7%, n = 3). On a typical day, eighty-four percent of participants (n = 38) indicated that they used public restrooms, 24% (n = 11) used a restroom in a private business, 22% (n = 10) used a portable restroom (i.e., porta potty), and 27% (n = 12) used the bushes, a bucket, cup, plastic bag, or other container. In response to the question, "Have you ever relieved yourself in a place not meant for it because you did not have access to a restroom facility?", most participants (78%, n = 35) indicated that they had relieved themselves in "a place not meant for it" (e.g.: a bucket, cup, bottle, or other container; a plastic bag/"flying toilet;" or the bushes) because they were unable to access a restroom.

Further, when asked "How often do you have to urinate/defecate outside due to a lack of public restrooms available?" (Table 5) over half (24, 53%) report that they do so all the time, often, or sometimes do this. Fifteen (33%) respondents often have to hold their urine during the day due to a lack of public restrooms available, and 10 (22%) sometimes have to "hold it." Over half of respondents (53%) report that their main restroom access (public or private), more often than not, has been limited by a facility's hours of operation or because of unsanitary conditions. This lack of access is complicated by multiple factors, including a public restroom being too far away or no public restrooms nearby (67%, n = 30), safety concerns (including it being too dark at night) (36%, n = 16), and being too tired to travel to a restroom

**Table 4. Sociodemographic characteristics of social survey participants experiencing homelessness in and around Pacific Beach (n = 45).**

| Sociodemographic Characteristics | Total (%) |
|---|---|
| **Gender** | |
| Male | 32 (71) |
| Female | 11 (24) |
| Non-binary | 1 (2) |
| Missing | 1 (2) |
| **Race/Ethnicity** | |
| White | 28 (62) |
| Black | 2 (4) |
| Hispanic/Latinx | 4 (9) |
| Multiracial/Multiethnic | 9 (20) |
| Other | 1 (2) |
| Missing | 1 (2) |
| **Sexual Orientation** | |
| Heterosexual | 41 (91) |
| Asexual | 1 (2) |
| Not sure / questioning | 1 (2) |
| Missing | 2 (4) |
| **Physical Disability** | |
| Yes | 19 (42) |
| No | 25 (56) |
| Missing | 1 (2) |
| **Main Source(s) of Income (current)[a]** | |
| Public Assistance (cash aid, food stamps, unemployment, disability, and/or social security insurance) | 25 (56) |
| Gainful employment (full-time, part-time, multiple jobs) | 8 (18) |
| Recycling | 7 (16) |
| Selling blood plasma | 1 (2) |
| Something else | 13 (29) |
| **Primary Sleeping Location (past 4 weeks)[a]** | |
| Vehicle | 16 (36) |
| Outdoors | 27 (60) |
| Temporary shelter or hotel room | 4 (9) |
| With family or friends | 2 (4) |
| Hospital | 1 (2) |
| Someplace else | 5 (11) |
| Missing | 1 (2) |

[a]Response options are not mutually exclusive.

Note: For mutually exclusive response options, percentages sum to 100%, except for rounding error.

facility (9%, n = 4). The easiest time for participants to access any restroom was in the morning (58%, n = 26), followed by afternoon (44%, n = 20), evening (40%, n = 18), and overnight (31%, n = 14); with 27% (n = 12) indicating that it is more difficult to find a restroom on the weekends.

Table 6 summarizes participant reports of the availability of supplies and amenities in public restrooms. While many features were always or often available, many participants noted that some of the most basic health and hygiene features were rarely or never available including toilet paper (n = 6), functioning sink (n = 5), soap (n = 14), hot water (n = 32), stall doors

**Table 5. Self-reported restroom access among participants experiencing homelessness in and around Pacific Beach (n = 45).**

| How often... | All the time/Often | Sometimes | Rarely/Never | Missing |
|---|---|---|---|---|
| Do you have to urinate/defecate outside due to a lack of public restrooms available | 14 (31) | 10 (22) | 19 (42) | 2 (4) |
| Do you have to hold your urine at any time of day due to a lack of public restrooms available? | 15 (33) | 10 (22) | 17 (38) | 3 (7) |
| Has your main restroom access been interrupted or limited (i.e. limited hours, too unclean to use)? | 10 (22) | 14 (31) | 17 (38) | 4 (9) |

(n = 4), and stall doors with functioning locks (n = 9). More than half of participants indicated that toilets/urinals were always or often functioning, however, a third indicated that toilets/urinals were only functioning sometimes.

When participants were asked to name supplies they often need in restrooms but that are rarely available, respondents most frequently mentioned soap (n = 22) and toilet paper (n = 14). Others mentioned needing paper towels (n = 8), hot water (n = 3), menstrual products (n = 3), and mirrors (n = 1).

## 3.4 Triangulation—Public restroom reliance, (in)accessibility, and restroom features

Although each of the tools described can be used separately, they were designed to be used together such that investigators and practitioners identify areas of consistency as well as relevant points of divergence across the results generated from each of the tools that may have remained hidden with the use of only one tool/method–referred to as "methods triangulation" [54,55]. Methods triangulation provides us the opportunity to use our tools, in conversation with one another, to corroborate findings, identify tensions and inconsistencies, and increase the validity of our findings [55]. We illustrate the utility of methods triangulation using the three tools in the case study and highlight below, the overlaps and differences in findings across each of the three tools.

Independently, each tool portrays public restroom accessibility slightly differently in PB, highlighting the need to consider macro- through micro-level perspectives, and the utility of

**Table 6. Reported availability of supplies and amenities in public restrooms (n = 45).**

| | All the time/Often | Sometimes | Rarely/Never | Missing |
|---|---|---|---|---|
| Toilet paper | 24 | 13 | 6 | 2 |
| Functioning sink | 27 | 11 | 5 | 2 |
| Soap | 13 | 15 | 14 | 3 |
| Hot water | 7 | 4 | 32 | 2 |
| Stall doors | 28 | 11 | 4 | 2 |
| Stall doors with functioning locks | 23 | 11 | 9 | 2 |
| Functioning toilets/urinals | 24 | 16 | 4 | 1 |
| Seat covers | 10 | 6 | 27 | 2 |
| Hand sanitizer | 3 | 3 | 37 | 2 |
| Something to dry your hands | 16 | 11 | 16 | 2 |
| Trash can | 37 | 2 | 4 | 2 |
| Well-lit | 22 | 12 | 9 | 2 |
| Menstrual products | 1 | 0 | 33 | 11 |
| Toothpaste/Toothbrushes | 4 | 2 | 37 | 2 |
| Condoms | 2 | 1 | 39 | 3 |
| Face masks | 2 | 2 | 39 | 2 |

Note: Participants responded to the prompt "Of the public restrooms that you use, how often are the following amenities available?".

using the three tools together. Focusing on the spatial database (Tool 1), PB appears to fare relatively better than other CPAs in San Diego with respect to public restroom counts. When used as the only tool, the spatial database can be combined with other macro-level information to understand population pressures on available restrooms.

Our macro-level analysis, which examined the spatial database in conjunction with neighborhood encampment data, indicated high likelihood of population pressures, for which we found additional support in the field assessment (Tool 2) and the social survey (Tool 3). Specifically, the field assessment and social survey both yielded different, yet converging evidence that public restrooms are maldistributed. The field assessment data indicated that accessibility to public restrooms may be limited due to their wide geographic dispersion, potential restrictions due to restroom type, and inconsistent hours of operation. Although objective data may indicate that PB has a higher than average number of public restrooms, subjective social survey data indicate that respondents often feel that restrooms are too far away when they need to use them (67%).

Furthermore, although we could confirm hours of operation for the two public building restrooms, hours of operation could not be confirmed for any of the outdoor public restrooms in PB during the field assessment; thus, the field assessment team was unable to discover true hours of all facilities in the PB region and characterize whether 24/7 access is available anywhere in the region. When we asked people about the easiest times of day to access restrooms, there was some evidence that overnight and weekend access is most difficult. Indeed, when asked the question, "In your opinion, what kinds of things would make it easier for you and the people you know to access basic hygiene and sanitation needs? (i.e. washing your hands, using the bathroom, and bathing/showering)", seven participants in the social survey mentioned expanding restroom hours on nights and/or weekends. Taken together, although the spatial data indicate that PB has a number of public restrooms that is above average relative to other CPAs, and that are generally distributed throughout the neighborhood, the field assessment and social survey data provide objective and subjective data indicating public restroom inaccessibility among unhoused people, a population that is highly public restroom reliant.

While a large majority of respondents continue to rely on public restrooms, data from the social survey add new evidence that the current provision of public restrooms is insufficient. Many participants indicated that they practiced open urination/defecation because they were unable to access a restroom. Many others reported having to "hold it" often because there was no restroom nearby and that restroom access has further been limited by restroom hours of operation and unsanitary conditions. While the total count of restrooms in PB is above average, social survey data provide us with evidence that access is still difficult. These individual-level reports of the impacts of insufficient restroom access highlight linkages to street-level health (open urination/defecation), individual-level health (not having anywhere to go to the bathroom), and environmental-health at restrooms (unsanitary conditions). Taken together, data generated by each of the tools emphasize the need for public restrooms to support public health.

It is important to note that the field assessment data were focused specifically on PB, whereas the social survey queried participants' experiences with public restrooms overall (not limited to PB). However, when layered together, the field assessment data may be used to comment on what restrooms in a region can provide to survey respondents who currently use the facilities in that region. The field assessments indicated that basic features such as soap and toilet paper were widely available in PB public restrooms. However, these field assessments were only collected at one point in time. In speaking to unhoused people in the social survey, many report that these basic features are not always available. For example, 46% of survey respondents indicated that toilet paper was not always available, and 71% reported that soap was not always available.

Social survey participants identified a number of restroom features that were "rarely/never" available in public restrooms, including hot water (71% reporting, n = 32). The field assessment found that two restrooms–those located in public buildings–had hot water available, but the remaining restrooms (which had hours that could not be confirmed) did not have hot water. Looking at the field assessment data on its own, this lack of hot water may seem innocuous. However, when layered with social survey data, it is evident that unhoused people who are reliant on public restrooms have very limited access to hot water and report that expanded access to hot water would make it easier to meet basic sanitation needs.

Each of our tools revealed new, sometimes divergent information on the availability of key features at PB restrooms. While the field assessment data indicated that PB restrooms were well-stocked, social survey respondents reporting on their lifetime history with public restrooms add nuance to our understanding of how restroom features may be highly variable. Social survey respondents provide us with restroom feature information that is otherwise hard to capture via cross-sectional field assessments at public restrooms.

## 4. Discussion

In this paper, we document both the methodological contributions of this work and highlight topical findings from a case study in San Diego, California. Together, these contributions provide new methods to investigate and understand public restroom availability and quality in a study region and individuals' experiences with restroom (in)access.

By using our tools in concert with one another, we are able to holistically document and assess access to and conditions of public restrooms in a given region, as well as the lived experiences and health needs related to restroom access of a population highly reliant on public restrooms (unhoused people). Together, the three tools extend prior research to offer layered insight into the true conditions of the sanitation landscape. The spatial database development provides a macro-level baseline understanding of the number and distribution of restrooms in the built environment. The tool offers a systematic approach to documenting public restroom availability where previous efforts have relied on incomplete administrative and crowd-sourced datasets. Restroom spatial database development is one approach to understanding gaps in public restroom availability and can serve to broaden public information on availability facilities. In addition, as demonstrated in the case study, the spatial database can be linked to other existing data to understand issues of sanitation inequity and the relationship between the availability of sanitation resources and health-related needs on a broad scale.

The field assessment tools provide a standardized approach to documenting conditions at individual restroom facilities. Field assessment tools can provide information on restroom feature availability that would be impossible with the spatial database on its own. By providing both an efficient (short-form) and detailed (long-form) method of characterizing health and hygiene features of public restrooms, the field assessment tool provides a reliable approach for researchers, practitioners, and advocates to document the extent to which facilities meet basic health and sanitation needs (e.g., to prevent infectious disease transmission), or to meet the needs of priority populations (e.g., unhoused people, people with small children, people who menstruate).

Finally, the social survey focuses on how restrooms are experienced by those who are most reliant on such public resources, as well as the relationship between restroom quantity, quality, and access and health-related needs and outcomes (e.g., inability to wash one's hands; holding urine due to lack of restroom availability) by collecting detailed information on users' perceptions and experiences of restroom availability and the features documented in the field assessment. Importantly, the social survey places spatial and field assessment data in the context of

users' needs and lived experiences, and can be tailored to capture the perspectives of specific subgroups who rely on public restrooms on a daily basis.

In the Pacific Beach case study, the spatial database (Tool 1) suggests a higher than average number of public restrooms in the region; however, four of the seven restrooms are clustered along the coastline with three located inland. With the addition of field assessment data (Tool 2), we determined that five restrooms were located in outdoor facilities and two in designated public buildings and as such, accessibility at these restrooms may be limited due to their geographic dispersion and hours of operation. This concern related to accessibility is affirmed by the social survey (Tool 3) where people experiencing homelessness report troubles using public restrooms because they are sometimes too far away and because of their limited hours. The social survey (Tool 3) also exposes the myriad health-threatening behaviors in response to this limited access, such as "holding it," using toilet alternatives like buckets, cups, plastic bags and other containers, and urinating and defecating in public, each of with have implications for negative health and social outcomes. For example, prior research finds that open defecation not only negatively affects physical health, but also psychosocial well-being, particularly among women [56], and that holding it may lead to urinary tract infections, incontinence, and kidney stones [10–12].

Tools 2 and 3 also highlight that even when restrooms are available, they often lack the most basic necessities, including toilet paper and soap–both of which are critical to preventing the spread of infectious diseases [57,58]. Thus, increasing restroom availability alone is unlikely to address the broader public health concerns affecting those who rely on public restrooms including disease transmission related to the inability to wash one's hands with soap and water.

Lastly, methods triangulation allowed us to corroborate findings, identify tensions and inconsistencies, and to increase the validity of our findings. In one critical example of this–hours of operation–our social survey respondents confirmed what we suspected but were unable to verify during neither our spatial database development nor our field assessment. More than half of survey respondents reported experiencing restricted access to their main restroom due to limited hours of operation and/or unsanitary conditions, with two-thirds of respondents reporting a public restroom being too far away or no public restrooms nearby where they stay (67%, n = 30). Further, while our academic research team's field assessment data suggested that PB restrooms were well-stocked, social survey respondents reporting on their lifetime history with public restrooms clarify that the provision of key resources like soap and toilet paper are in fact highly variable. Also, hot water emerged as a critical and valued resource for the unhoused population that was often absent at public restrooms.

In addition to their strengths, it is important to note limitations of the tools described here and consider opportunities for future research. First, the spatial database development relied on data from several sources. While these data were verified in numerous ways, it is still possible that public restrooms, or restrooms perceived by the public to be "public," may be left out. However, a strength of our approach is the application of a standard definition for "public restroom" with clear inclusion/exclusion criteria that allowed us to combine data sources, engage in a verification process, and train assessors for the spatial database development. This approach could be scaled up to larger geographies (entire states or countries) for development of more comprehensive public sanitation resources maps. Second, while we continue to assess restrooms across San Diego, we know that the assessment is largely a snapshot. The current study lacks a re-assessment protocol for restrooms. To get a better sense of restroom features, we would need to assess restrooms at multiple timepoints and at different days/times of the week. Future research using this tool may benefit from repeat assessments for inter-assessment reliability modeling. Finally, in administering the social survey, we asked participants to

comment on their general experiences with public restrooms; however, there is a chance these restrooms differ from those mapped and/or assessed in the same study area (for example, the restrooms referenced by participants may be not considered public or may be in a different region of a City of County. Future research could make this connection to specific facilities more explicit by changing the survey instrument to ask about specific restrooms, such as those along transit lines [59] or researchers could conduct go-along interviews with participants at local restrooms [60]. In addition, future research could leverage photovoice and other audio-visual participatory methods to document community needs and drive collective action related to sanitation justice and public restroom access [61]. This would allow for direct comparisons between the field assessment data and survey respondents' impression of and experiences with the restroom.

To conclude, this paper presents three tools for examining sanitation justice from the macro- to the micro-level including spatial database development, field assessment of public restrooms, and a social survey of unhoused people regarding their experiences with public restrooms. This suite of methods gives us new and layered information about public restroom availability, quality, and user experiences, expanding our ability to approach research on public restrooms and public health.

## Supporting information

**S1 Appendix.**
(DOCX)

## Acknowledgments

The authors of this article would like to thank Think Dignity staff and leadership: Mitchelle Woodson, Executive Director, Merlynn Watanabe, Programs and Operations Manager, and Christine Lopez, Community Engagement Coordinator for their insight and advocacy related to this work. We would also like to thank Adriana K. Rios, Nicolas Gutierrez III, and Victor Maisano for their assistance with maps and figures.

## Author Contributions

**Conceptualization:** Madison R. E. Swayne, Jerel P. Calzo, Jennifer K. Felner, Megan Welsh Carroll.

**Data curation:** Madison R. E. Swayne, Jerel P. Calzo, Jennifer K. Felner, Megan Welsh Carroll.

**Formal analysis:** Madison R. E. Swayne, Jerel P. Calzo, Jennifer K. Felner, Megan Welsh Carroll.

**Funding acquisition:** Megan Welsh Carroll.

**Investigation:** Madison R. E. Swayne, Jerel P. Calzo, Jennifer K. Felner, Megan Welsh Carroll.

**Methodology:** Madison R. E. Swayne, Jerel P. Calzo, Jennifer K. Felner, Megan Welsh Carroll.

**Project administration:** Megan Welsh Carroll.

**Software:** Madison R. E. Swayne.

**Supervision:** Megan Welsh Carroll.

**Visualization:** Madison R. E. Swayne.

**Writing – original draft:** Madison R. E. Swayne, Jerel P. Calzo, Jennifer K. Felner, Megan Welsh Carroll.

**Writing – review & editing:** Madison R. E. Swayne, Jerel P. Calzo, Jennifer K. Felner, Megan Welsh Carroll.

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
