## [Decision Letter · Decision Letter 0]

2 May 2023

PONE-D-23-02687Developing evidence for building sanitation justice: A multi methods approach to understanding public restroom quantity, quality, accessibility, and user experiencesPLOS ONE

Dear Dr. Swayne,

Thank you for submitting your manuscript to PLOS ONE. After careful consideration, we feel that it has merit but does not fully meet PLOS ONE’s publication criteria as it currently stands. Therefore, we invite you to submit a revised version of the manuscript that addresses the points raised during the review process.

We look forward to receiving your revised manuscript.

Kind regards,

Ranjit Kumar Dehury

Academic Editor

PLOS ONE

Journal Requirements:

2. In the ethics statement in the Methods, you have specified that verbal consent was obtained. Please provide additional details regarding how this consent was documented and witnessed, and state whether this was approved by the IRB.

"The authors received support from the San Diego State University (SDSU) Emergency Funding for Student Assistance with Research, Scholarship, and Creative Activities (RSCA) Award (2021); the SDSU Summer Undergraduate Research Program (2021, 2022); SDSU’s Rapid Response Grants program; SDSU’s Seed Grant program; and SDSU’s School of Public Affairs. No grant numbers available. "

6. We note that Figures 3 and 4 in your submission contain [map/satellite] images which may be copyrighted. All PLOS content is published under the Creative Commons Attribution License (CC BY 4.0), which means that the manuscript, images, and Supporting Information files will be freely available online, and any third party is permitted to access, download, copy, distribute, and use these materials in any way, even commercially, with proper attribution. For these reasons, we cannot publish previously copyrighted maps or satellite images created using proprietary data, such as Google software (Google Maps, Street View, and Earth). For more information, see our copyright guidelines: http://journals.plos.org/plosone/s/licenses-and-copyright.

a. You may seek permission from the original copyright holder of Figures 3 and 4 to publish the content specifically under the CC BY 4.0 license.  

Additional Editor Comments:

The author may kindly do a through revision by incorporating the comments of reviewers.

Reviewers' comments:

Reviewer's Responses to Questions

**Comments to the Author**

1. Is the manuscript technically sound, and do the data support the conclusions?

Reviewer #1: Yes

Reviewer #2: Yes

2. Has the statistical analysis been performed appropriately and rigorously? 

Reviewer #1: N/A

Reviewer #2: Yes

3. Have the authors made all data underlying the findings in their manuscript fully available?

Reviewer #1: No

Reviewer #2: Yes

4. Is the manuscript presented in an intelligible fashion and written in standard English?

Reviewer #1: Yes

Reviewer #2: Yes

5. Review Comments to the Author

Reviewer #1: The manuscript has focused on a serious issue on Public Health and Hygine, at San Diego, US. Authors have described that they have developed three tools, "public restroom spatial database, field assessment tool, and

survey, and document process to collect these public restroom data in a

consistent, public health and equity-driven framework". Although the strategy of the tool development has been expalined in the manuscript, it has not explained the languages/codes used for tool and database development. In method section, Tool 1, page 179, there is a mention about 669 datapoints (public restrooms). This data is converted into a map and made public on ArcGIS online. The ArcGIS online URL is missing. Not sure whether the 669 datapoints (raw data) has been shown in the above-mentioned website.

The main source of data repoistory (https://bathrooms.sdsu.edu/) is not mentioned explicitly in the manuscript text or abstract. Only mentioned in the last but second figure, (that is also with poor resolution). Without highlighting this data source, explicity, it is diffcult to understand the consequences of the results.

There are four figures mentione in the text. However, those are very difficult to trace, because those four figures attached at the end of the mansucript do not contain any labels (although Figure legends are mentioned within the text).

Moreover, the qualities (resolutions) of the figures are very poor. The texts embedded in the figures, specifically, the last two are almost illegible.

Overall the work addressed a serious public issue worth to be considered and put forward for action, it requires minor changes before in can be considered for acceptance.

Reviewer #2: This article aims to address the very critical issues related to sanitation security among PEH group in San Diego. The methods undertaken and analysis part is also appropriate. But, the word limit is seems to be more and presentation style requires more concise, clear meaning to the reader and requires major revision. There are typo and punctuation error and formatting correction is required ( for example Sentence 71 is not completed; Sentence 81, more explanation the meaning of e.g. Flush-Toilet Finder (22) and Toilet Finder (23) is required etc). The limitation of the study should be clearly highlighted. The IRB approval for the study must be taken.

Since there are three methodological tool has been developed for sanitation justice, all technical concepts related to methodological development must be addressed. Also detailed description of data triangulation for case study approach needs to be highlighted. The key findings in the conclusion section need to be tightly built up.

Hope the reviewer comments would be helpful to strengthen the manuscript.

6. PLOS authors have the option to publish the peer review history of their article (what does this mean?). If published, this will include your full peer review and any attached files.

Reviewer #1: No

Reviewer #2: No

---

## [Author Response · Author response to Decision Letter 0]

25 May 2023

We have provided responses to each of the reviewer and editor comments in our Response to Reviewers letter that is attached to this submission.

---

## [Editor Report · Decision Letter 1]

28 Jun 2023

Developing evidence for building sanitation justice: A multi methods approach to understanding public restroom quantity, quality, accessibility, and user experiences

PONE-D-23-02687R1

Dear Dr. Madison Swayne,

We’re pleased to inform you that your manuscript has been judged scientifically suitable for publication and will be formally accepted for publication once it meets all outstanding technical requirements.

Kind regards,

Ranjit Kumar Dehury

Academic Editor

PLOS ONE

Additional Editor Comments (optional):

Dear authors,

The paper need fine tuning before being published in the journal. The auhours have incorporated the points raised by the reviewers.

With regards,

Ranjit
---

## [Editor Report · Acceptance letter]

6 Jul 2023

PONE-D-23-02687R1 

Developing evidence for building sanitation justice: A multi methods approach to understanding public restroom quantity, quality, accessibility, and user experiences 

Dear Dr. Swayne:

I'm pleased to inform you that your manuscript has been deemed suitable for publication in PLOS ONE. Congratulations! Your manuscript is now with our production department. 

Kind regards, 

on behalf of

Dr. Ranjit Kumar Dehury 

Academic Editor

PLOS ONE